# Freely Available, Fully Automated AI-Based Analysis of Primary Tumour and Metastases of Prostate Cancer in Whole-Body [^18^F]-PSMA-1007 PET-CT

**DOI:** 10.3390/diagnostics12092101

**Published:** 2022-08-30

**Authors:** Elin Trägårdh, Olof Enqvist, Johannes Ulén, Jonas Jögi, Ulrika Bitzén, Fredrik Hedeer, Kristian Valind, Sabine Garpered, Erland Hvittfeldt, Pablo Borrelli, Lars Edenbrandt

**Affiliations:** 1Department of Translational Medicine, Wallenberg Centre for Molecular Medicine, Lund University, 205 02 Malmö, Sweden; 2Department of Clinical Physiology and Nuclear Medicine, Skåne University Hospital, 205 02 Malmö, Sweden; 3Eigenvision AB, 211 30 Malmö, Sweden; 4Department of Electrical Engineering, Chalmers University of Technology, 412 96 Gothenburg, Sweden; 5Department of Clinical Physiology and Nuclear Medicine, Skåne University Hospital, 221 85 Lund, Sweden; 6Department of Clinical Physiology, Region Västra Götaland, Sahlgrenska University Hospital, 413 45 Gothenburg, Sweden; 7Department of Molecular and Clinical Medicine, Institute of Medicine, Sahlgrenska Academy, University of Gothenburg, 405 30 Gothenburg, Sweden

**Keywords:** deep learning, convolutional neural network, PSMA, artificial intelligence, prostate cancer

## Abstract

Here, we aimed to develop and validate a fully automated artificial intelligence (AI)-based method for the detection and quantification of suspected prostate tumour/local recurrence, lymph node metastases, and bone metastases from [^18^F]PSMA-1007 positron emission tomography-computed tomography (PET-CT) images. Images from 660 patients were included. Segmentations by one expert reader were ground truth. A convolutional neural network (CNN) was developed and trained on a training set, and the performance was tested on a separate test set of 120 patients. The AI method was compared with manual segmentations performed by several nuclear medicine physicians. Assessment of tumour burden (total lesion volume (TLV) and total lesion uptake (TLU)) was performed. The sensitivity of the AI method was, on average, 79% for detecting prostate tumour/recurrence, 79% for lymph node metastases, and 62% for bone metastases. On average, nuclear medicine physicians’ corresponding sensitivities were 78%, 78%, and 59%, respectively. The correlations of TLV and TLU between AI and nuclear medicine physicians were all statistically significant and ranged from R = 0.53 to R = 0.83. In conclusion, the development of an AI-based method for prostate cancer detection with sensitivity on par with nuclear medicine physicians was possible. The developed AI tool is freely available for researchers.

## 1. Introduction

Prostate cancer is one of the most diagnosed cancer types and one of the most common causes of cancer-related deaths among men worldwide [1]. Correct staging and the discovery of sites of recurrence are of utmost importance in making an informed treatment decision. One of the emerging methods for initial staging of patients with high-risk prostate cancer and for identifying sites of recurrent disease is prostate-specific membrane antigen-ligand (PSMA) positron emission tomography with computed tomography (PET-CT). PSMA is a transmembrane protein that is significantly overexpressed in malignant prostate tissue [2]. The method is more sensitive and accurate than the conventional imaging standard of CT and bone scan [3,4,5,6,7,8]. Different PSMA radiopharmaceuticals exist, such as, for example, [^68^Ga]PSMA-11 and [^18^F]PSMA-1007 [7,9]. The interpretation relies on visual inspection and is subject to inter- and intra-observer variability. The number of PET-CT examinations has increased substantially in recent years, especially for PSMA PET-CT. There is currently a deficit of nuclear medicine physicians and radiologists able to interpret the studies, and this gap is further widening with the actual trend. Artificial intelligence (AI) can help with both standardisations of interpretation and acting as a second opinion to nuclear medicine physicians. A third possible use of AI could be to quantify PSMA-positive tumour burden to decide on [^177^Lu]PSMA treatment and to evaluate the response of the treatment on consecutive PSMA PET-CT examinations. Tumour burden has been shown to correlate with overall survival in patients with advanced prostate cancer [10,11,12,13,14].

This study aimed to develop and validate a fully automated AI-based method for detecting and quantifying suspected prostate tumour or local recurrence, lymph node metastases, and bone metastases in [^18^F]PSMA PET-CT in patients with newly diagnosed high-risk prostate cancer and in patients with recurrent disease. Model performance was assessed on a lesion level, and the assessment of tumour burden was calculated from the automated segmentations. The AI-based method was compared with nuclear medicine physicians. A secondary aim was to make the AI-based tool freely available to other researchers.

## 2. Materials and Methods

### 2.1. Patients and Imaging

A total of 660 patients referred for clinically indicated [^18^F]PSMA PET-CT at Skåne University Hospital, Lund and Malmö, Sweden, from December 2019 to December 2020, were included, either due to initial staging of high-risk prostate cancer or for the detection of sites of suspected recurrent disease. The patients were injected with 4 MBq/kg of [^18^F]PSMA-1007. After a 2-h accumulation time, images were acquired on a Discovery MI PET-CT (GE Healthcare, Milwaukee, WI, USA) from the base of the skull to the mid-thigh.

The PET acquisition time was 2 min/bed position. The images were reconstructed using a block-sequential regularisation expectation maximisation algorithm (Q.Clear; GE Healthcare, Milwaukee, WI, USA) and a beta factor of 800 [15]. Time-of-flight, point spread function modelling, and a 256 × 256 matrix (pixel size 2.7 × 2.7 mm^2^, slice thickness 2.8 mm) were used.

A diagnostic CT with oral and intravenous contrast was performed and used for attenuation correction and anatomic correlation. The CT used for attenuation correction was acquired in the late venous phase. Tube current modulation was applied by adjusting the tube current. The noise index was 37.5, the tube voltage was 100 kV, and the slice thickness was 0.625 mm. An adaptive statistical iterative reconstruction technique was applied. This study was approved by the local research ethics committee at Lund University (#2016/417, #2018/753 and #2021–05734-02), and followed the principles of the Declaration of Helsinki. The patients provided written informed consent.

### 2.2. Manual Segmentations for Training (Ground Truth)

One experienced nuclear medicine physician segmented suspected prostate tumour or local recurrence, intra- and extra-pelvic lymph node metastases, and suspected bone metastases in the [^18^F]PSMA PET-CT images (denoted as Reading A, see below). The cloud-based annotation platform RECOMIA (https://www.recomia.org) was used for the manual segmentations and included basic display features for PET-CT images and segmentation tools [16]. From the full set, 120 studies were used as a test set. The remaining 540 images were divided into a training set (420 studies) and a validation set (120 images).

### 2.3. AI Tool

The model consists of a Unet3D CNN [17] trained to classify each pixel as either prostate tumour or recurrence, lymph node metastases, bone metastases, or background. The CNN has three inputs, the CT image (clamped to [−800, 800 HU] and normalised), the PET image (clamped to [0, 25] and normalised), and a multi-channel organ mask; see Appendix A. The organ mask is automatically created using only the CT image and the method described in [16]. All input images are rescaled to 1.37 mm × 1.37 mm × 2.79 mm per pixel.

### 2.4. Sampling

The model was trained using image patches, subsets of the full image. Since most of the image is background, how the patches are chosen is important. For each image, a sample mask is created with a weight for each pixel, encoding how often patches should be sampled with this pixel as a centre. The training is initialised with a sample mask created such that 50% of the patches are background and 50% are foreground, where each foreground label is equally likely to be chosen.

### 2.5. Training

The network is trained using two dropout layers with a dropout rate of 0.25 and l2 weight regularisation with a weight of 0.001. Categorical cross-entropy is used as a loss function, where foreground pixels are given 2.5-times-larger weight than background pixels, favouring sensitivity over precision. The loss is optimised using the Adam method [18] with Nesterov momentum and an initial learning rate of 0.001. The learning rate is cut in half if the validation loss has not decreased for five epochs. The input patches are augmented using a scaling of −10 to 10%, rotation of −0.15 to 0.15 radians, and intensity shifts of −0.5 to 0.5 for the PET image and −100 to 100 HU for the CT image.

Each epoch uses 20,000 samples for training and 10,000 samples for validation; the model is trained for ten epochs. After this half of the training, images are randomly selected, and the loss is calculated for each pixel in the selected images. The sampling masks for the randomly selected images are then updated. The updated weight $w_{p}^{i+1}$ for pixel *p* is:(1)$$w_{p}^{i+1}=0.5w_{p}^{i}+0.5\frac{\max_{p\in P}w_{p}^{i}}{\max_{p\in P}l_{p}}l_{p}$$
where $w_{p}^{i}$ is the old sample weight for pixel *p*, *l_p_* is the loss for pixel *p*, and *P* is the set of all pixels. This procedure will influence the sampling to sample regions with high loss more often as repeated 5 times.

### 2.6. Model Evaluation

The performance of the AI-based method was assessed by the test set of 120 patients, using 3 sets of “expert readers”. One of the expert readers (Reading A) was an experienced nuclear medicine physician (10 years of PET-CT experience) who also performed the manual segmentations for the model training. Six other nuclear medicine physicians (four with >10 years of PET-CT experience and two with 5–10 years of PET-CT experience) segmented suspected tumours/metastases in 40 cases each from the test set of 120 patients. Thus, each PET-CT scan in the test set was segmented by two expert readers who were not involved in the model training. The physicians segmented cases randomly, yielding two separate sets of segmentations for each patient, referred to as Reading B and Reading C below.

The instructions to all readers were to: (1)Segment the suspected malignant lesions in the prostate and seminal vesicles in patients with prostate and in patients without a prostate (after prostatectomy) to segment any suspicious recurrence in the prostate bed and/or seminal vesicles.(2)To mark all suspected lymph node metastases, both pelvic and extra-pelvic, using the E-PSMA grading system as guidance [19]. Low uptake (equal to or lower than the background) in lymph nodes was generally considered non-pathologic while intense uptake (above liver) was marked pathologic. Intermediate uptake was generally considered pathologic when deviating from known patterns of unspecific uptake (such as low-intermediate uptake along the distal external iliac vessels or in the mediastinum). If highly suspicious lymph nodes with low uptake were found, for example, enlarged necrotic pelvic lymph nodes, the physicians were instructed to segment these as suspected lymph node metastases.(3)To mark uptakes in the bone that could be metastases but, in general, not low-grade uptake in, for example, the ribs and pelvic bones, which is a common unspecific finding in [^18^F]PSMA-1007.

The model was evaluated on a lesion-based level, defined as the model’s ability to detect the tumours/metastases identified by one of the three readings. The locations assessed were suspected prostate tumour/recurrence, suspected lymph node metastases, and suspected bone metastases. To assess inter-reader variability, the expert readers were also compared. Readings A, B, and C were alternately used as a reference and pairwise compared to either another reading or AI. True positive lesions for a human reading or AI were defined as either partial or full segmentation overlap with another reading used as a reference; otherwise, they were considered false negative. Lesions detected by a human reading or AI without segmentation overlap with the reading used as ground truth were regarded a false positive. The sensitivity was calculated as the proportion of suspected lymph node metastases detected by a reading or AI out of those detected by a reading used as ground truth. The positive predictive value (PPV) was calculated as the proportion of true positive lesions for human reading or AI compared to a reference reading, divided by false positive plus true positive lesions compared to the same reference reading.

Tumour burden was also assessed for prostate tumour/recurrence, suspected lymph node metastases, and suspected bone metastases by measuring the total lesion volume (TLV) and the total lesion uptake (TLU). TLV was calculated by adding the volume of all positive voxels identified in the automated or manual segmentations, and TLU was calculated by summing the SUVmean to the TLV for each lesion and then summing all lesion TLUs to a total TLU.

### 2.7. Statistical Analysis

Sensitivity and PPV were assessed for the AI and the different readings described above. Correlations between the tumour burden (TLV and TLU) measured by the AI and Reading A were assessed by Spearman rank correlation. A significance level of *p* = 0.05 was used.

## 3. Results

### 3.1. Detection of Suspected Tumours and Metastases

On a lesion level, the AI model had an average sensitivity of 79% (range, 70–95%) for detecting suspected tumour(s) in the prostate or local recurrence. When each reading was alternately used as a reference and tested against the other readings, the average sensitivity was 78% (range, 63–94%). For detecting suspected lymph node metastases, the AI model had an average sensitivity of 79% (range 69–88%), whereas the human readings had a sensitivity of 78% (range, 63–94%). Finally, the average sensitivity for detecting suspected bone metastases was 62% (range, 45–89%) for the AI and 59% (range, 29–87%) for the human readings (Figure 1).

The highest sensitivities for the AI method were obtained when Reading A (whose segmentations were used for training the AI) was used as a reference (95%, 88%, and 89% for prostate, lymph node, and bone lesions, respectively).

Table 1, Table 2 and Table 3 show the number of true positive, false positive, and false negative lesions for a prostate tumour or local recurrence, lymph node metastases, and bone metastases, respectively. As shown in the tables, the number of false negatives per patient was rather low for both “AI vs. Reading” and “Reading vs. Reading” for all tumour/metastases locations. A higher number of false positive lesions were found for the comparison “AI vs. Reading” for lymph node metastases and bone metastases compared with “Reading vs. Reading”.

### 3.2. Quantification of Tumour Burden

The tumour burden, measured as TLV and TLU, was rather similar between the AI and the three different readings, where the TLU and TLV ranges for the AI were within the ranges of the different readings except for TLU for lymph node metastases (Figure 2 and Table 4). The correlations for TLV and TLU between AI and Reading A were all statistically significant and ranged from moderate (R = 0.53 for TLV lymph node metastases) to strong (R = 0.83 for TLU prostate tumour/recurrence) (Figure 3). The prostate tumour/recurrence results were clearly influenced by one outlier with a high tumour burden. Figure 4 shows a patient example for the detection and quantification of suspected pathology.

## 4. Discussion

We developed and validated a fully automated AI-based method for detecting prostate cancer tumours and metastases and quantifying tumour burden in [^18^F]PSMA-1007 PET-CT scans. We found a relatively high sensitivity within the range of nuclear medicine physicians and a significant correlation for tumour burden assessment (TLV and TLU) between the AI and manual segmentations.

Automated methods for the detection of tumour/metastases and quantification of tumour burden could have several clinical implications. First, it could help to decrease the inter-individual differences in image interpretation. This study shows that the inter-individual differences when physicians are forced to dichotomise findings into benign or malignant are substantial, even among experienced readers guided by E-PSMA interpretation criteria. Automated image interpretation has decreased inter-individual differences in, for example, bone scan readings [20]. Second, it could help speed up the interpretation time, which is important when nuclear medicine physicians/radiologists are scarce. Third, a validated and fast assessment of tumour burden could have several implications, for example, providing prognostic information [10,11,12] and evaluating treatment response [21,22].

In this study, we found a much better overall performance for the AI model when Reading A was used as the reference (whose segmentations acted as ground truth) than when other readings were used as the reference. It is hardly surprising that the AI resembles the segmentations performed by the expert reader that was used for training the network. This, however, emphasises the importance of having a robust ground truth. Ideally, the ground truth should be confirmed by, for example, histopathology, which unfortunately is not feasible for all suspected lesions.

We found lower sensitivities for the detection of bone metastases and between manual readings. It is well-known that [^18^F]PSMA-1007 has increased tracer uptake in benign bone lesions in addition to bone metastases. Unspecific bone uptakes have previously been found in at least 50% of patients, most commonly in the ribs and the pelvis [23,24]. The unspecific uptakes were also more frequent in images obtained with a digital PET-CT scanner, which was used in our study [23]. The particularly low sensitivity between AI and manual readings using Readings B and C as references could be attributed to differences in how the different expert readers handled focal uptakes in bones, where many could be suspected to be unspecific.

We have previously published articles on AI-based detection and quantification of PET-CT images in patients with prostate cancer [12,13,25,26,27,28] using different radiopharmaceuticals. This study is a development of one of our most recently published studies [28], which also aimed to develop an AI-based method for [^18^F]PSMA-1007 PET-CT images but only included patients referred for initial staging, and the AI method was only trained for the detection of suspected pelvic lymph node metastases. Furthermore, it only included 161 PET-CT scans for training (540 in the present study). A recent paper by Johnsson et al. [29] found just over 90% sensitivity for automated detection of lymph node metastases for [^18^F]DCFPyL PET-CT scans and 87% for bone lesions. However, they had a larger number of false-positive lesions per patient (19.5 per patient for pelvic lymph node metastases, 90.5 for all lymph node metastases, and 8.2 for bone lesions). The numbers in the present study were 2.8 for lymph node metastases and 3.0 for bone metastases.

The rate of false positives and false negatives deserves further comment. If an AI-method is used for highlighting possible malignant lesions to be evaluated by a physician, it is important to keep the number of false negative lesions as low as possible. However, if the number of false positive lesions is very high, it will be time-consuming for the physician to consider and discard lesions marked by the AI. If the AI is used to quantify, for example, the tumour burden, the rate of false positives and false negatives should be balanced. When training the AI, it is possible to bias the networks towards either a high sensitivity (at the cost of many false positives) or having a high PPV (at the cost of many false negatives). In this study, we aimed to keep the number of false negatives relatively low at the expense of a slightly higher false positive rate.

Another emerging research field, closely related to this work, is radiomics. Through AI, it may be possible to automatically extract quantitative features in medical images, which can be used, for example, for risk stratification and response prediction. Most radiomics studies on prostate cancer are performed on magnetic resonance imaging, but some PSMA PET-CT studies exist [30,31]. When validated imaging biomarkers are available, this could be an important part of personalized medicine.

The results should be viewed in light of some limitations. All PET-CT scans were from a single institution, leading to selection bias in the patient material. It is also unknown how the AI algorithm performs for images obtained with different PET-CT protocols, including different accumulation times, PET-CT scanners, and reconstruction algorithms. The manual segmentations used as ground truth cannot be considered perfect. Ideally, histopathologic verification should be carried out, but it is often not feasible. The patients included in this study were biased toward a low disease burden. The model should also be validated in patients with a higher tumour burden. The method is trained on [^18^F]PSMA-1007 PET-CT images and does not necessarily perform equally well if other PSMA radiopharmaceuticals are used.

## 5. Conclusions

In this study, we showed that it was possible to develop an AI method with sensitivity on par with nuclear medicine physicians to detect prostate tumours or local recurrence, lymph node metastases, and bone metastases in [^18^F]PSMA-1007 PET-CT scans. Tumour burden assessment by AI was moderate to strongly correlated to that of an expert reader. The difficulty in achieving high inter-observer reproducibility emphasises the need for automated methods. To further validate the method, we made our AI tool freely available to other researchers at www.recomia.org or by emailing contact@recomia.org.

## Figures and Tables

**Figure 1 diagnostics-12-02101-f001:**
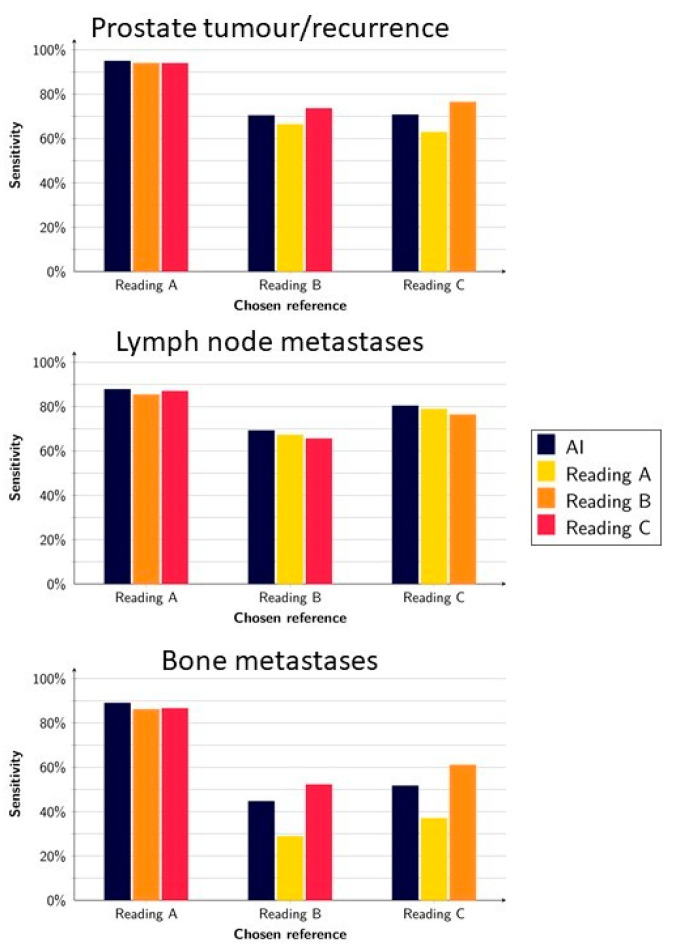
Sensitivity of the AI model and readings when using Reading A, B, and C as a reference, respectively.

**Figure 2 diagnostics-12-02101-f002:**
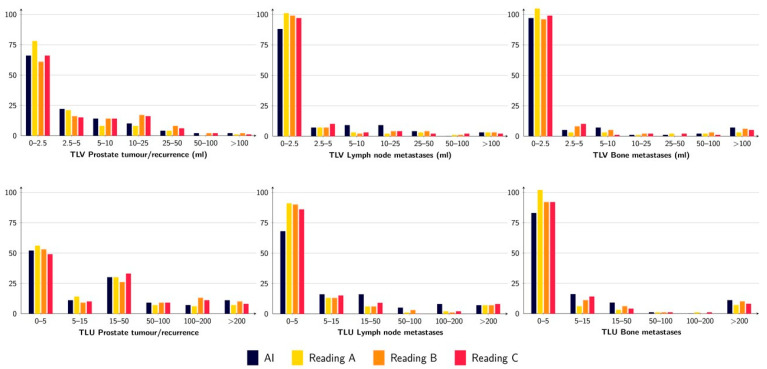
TLV (**upper row**) and TLU (**lower row**) for prostate tumour/recurrence (**left**), lymph node metastases (**middle**), and bone metastases (**right**) for the AI and the three different readings. The figure shows the number of patients with a certain range of measured TLV/TLU.

**Figure 3 diagnostics-12-02101-f003:**
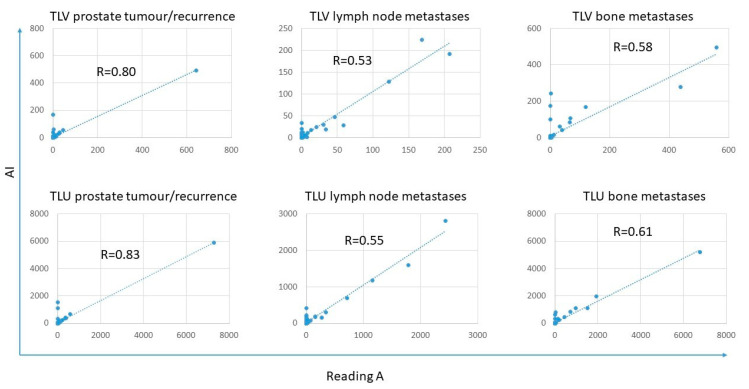
Correlations between TLV and TLU were obtained for prostate tumour/recurrence, lymph node metastases, and bone metastases between Reading A and AI.

**Figure 4 diagnostics-12-02101-f004:**
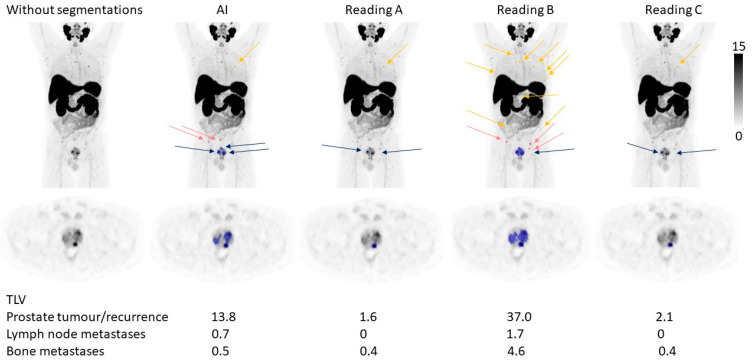
A patient example. The arrows indicate the number and locations of suspected tumours in the prostate (blue), lymph node metastases (pink), and bone metastases (yellow) detected by the AI and different manual readings. Below the figure is the TLV for the different tumour locations noted.

**Table 1 diagnostics-12-02101-t001:** True and false positives (TP/FP), false negatives (FN), sensitivity, and positive predictive value (PPV) for the detection of **prostate tumour/recurrence**. The numbers show the average and range when one reader at a time was used as the reference.

*n* = 120 Patients	AI vs. Reading	Reading vs. Reading
TP (*n*)		
-Total	93.7 (88–99)	92.7 (83–107)
-Per patient	0.8 (0.7–0.8)	0.8 (0.7–0.9)
FP (*n*)		
-Total	62.3 (54–77)	28.7 (6–52)
-Per patient	0.5 (0.4–0.6)	0.2 (0.05–0.4)
FN (*n*)		
-Total	27.7 (5–41)	28.7 (6–52)
-Per patient	0.2 (0.04–0.3)	0.2 (0.05–0.4)
Sensitivity (%)	78.7 (70.4–94.9)	77.9 (62.9–93.9)
PPV (%)	60.3 (55.0–63.9)	78.3 (64.1–93.6)

**Table 2 diagnostics-12-02101-t002:** True and false positives (TP/FP), false negatives (FN), sensitivity, and positive predictive value (PPV) for the detection of **lymph node metastases**. The numbers show the average and range when one reader at a time was used as the reference.

*n* = 120 Patients	AI vs. Reading	Reading vs. Reading
TP (*n*)		
-Total	215.3 (209–221)	208.7 (198–217)
-Per patient	1.8 (1.7–1.8)	1.7 (1.7–1.8)
FP (*n*)		
-Total	333.3 (331–335)	65.7 (32–104)
-Per patient	2.8 (2.8–2.8)	0.5 (0.3–0.9)
FN (*n*)		
-Total	59.0 (30–93)	65.7 (32–104)
-Per patient	0.5 (0.3–0.8)	0.5 (0.3–0.9)
Sensitivity (%)	79.1 (69.2–87.8)	77.9 (65.6–87.0)
PPV (%)	39.2 (38.5–40.0)	78.3 (66.9–87.1)

**Table 3 diagnostics-12-02101-t003:** True and false positives (TP/FP), false negatives (FN), sensitivity, and positive predictive value (PPV) for the detection of **bone metastases**. The numbers show the average and range when one reader at a time was used as the reference.

*n* = 120 Patients	AI vs. Reading	Reading vs. Reading
TP (*n*)		
-Total	236.7 (186–271)	222.2 (175–317)
-Per patient	2.0 (1.6–2.3)	1.9 (1.5–2.6)
FP (*n*)		
-Total	357.7 (260–451)	213.2 (28–432)
-Per patient	3.0 (2.2–3.8)	1.8 (0.2–3.6)
FN (*n*)		
-Total	198.7 (23–336)	213.2 (29–432)
-Per patient	1.7 (0.2–2.8)	1.8 (0.2–3.6)
Sensitivity (%)	61.8 (44.6–89.0)	58.6 (28.8–86.6)
PPV (%)	40.5 (29.2–51.0)	58.7 (29.4–86.6)

**Table 4 diagnostics-12-02101-t004:** Median (range) values of tumour burden, measured as TLV and TLU for the prostate tumour/recurrence, lymph node metastases, and bone metastases for AI and the readings.

	AI	Reading A	Reading B	Reading C
TLV Prostate tumour	1.6 (0–491)	1.2 (0–641)	1.9 (0–2749)	2.1 (0–640)
TLV Lymph nodes metastases	0.8 (0–224)	0 (0–207)	0 (0–336)	0 (0–174)
TLV Bone metastases	0.5 (0–495)	0 (0–559)	0 (0–1398)	0 (0–669)
TLU Prostate tumour	11.3 (0–5903)	6.4 (0–7291)	12.4 (0–8007)	12.8 (0–7326)
TLU Lymph nodes metastases	4.0 (0–2810)	0 (0–2433)	0 (0–2382)	0 (0–2390)
TLU Bone metastases	1.5 (0–5199)	0 (0–6763)	0 (0–7389)	0 (0–6570)

## Data Availability

The AI method is freely available for research at www.recomia.org.

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
