# Peer review of "Freely Available, Fully Automated AI-Based Analysis of Primary Tumour and Metastases of Prostate Cancer in Whole-Body [18F]-PSMA-1007 PET-CT"

_diagnostics, 2022, doi:10.3390/diagnostics12092101_

Round 1

Reviewer 1 Report

This paper reflects a most important subject concerned with AI applications  in the area of clinical evaluation of PET/CT images. This will be very important in the future as an aid to clinical imaging diagnosis. I find no flaws in the statement of the objectives, in the methodology or in the results. I think the future free availability of the tool is very meritorious. Congratulations for this work.

Author Response

We thank the reviewer for this kind statement.

Reviewer 2 Report

Radiomics represents one of the most promising fields of cancer research. It is now possible, through artificial intelligence (AI) and machine learning (ML) advanced algorithms, to extract abundant quantitative features from patients scans and to analyze the high amount of data coming from these novel diagnostic tools to ultimately improve the non-invasive risk stratification and disease management of patients with prostate cancer (PCa).

This study aims to develop and validate a fully automated artificial intelligence (AI)-based method for the detection and quantification of suspected prostate tumor/local recurrence, lymph node and bone metastases from [18F]PSMA-1007 positron emission tomography-computed tomography (PET-CT) images.

I believe that the study has sufficient merit to be considered for publication, although major revisions are required. Indeed, there are only a few spelling mistakes: authors should check them for clarity. However, tables and graphics are clearly described, but it is lacking in some points that would add value to the entire manuscript:

The authors should provide more information about radiomic, underlying status and current evidence of PSMA-PET in evaluation and follow-up of PCa. Here are two up-to-date articles from which to draw the necessary and correct information. (https://doi.org/10.1177%2F17562872221109020, https://doi.org/10.3390%2Fijms22189971) 

Author Response

We would like to thank the reviewer for the thorough reading of this manuscript and for the constructive suggestions for improving the manuscript.

We have added a paragraph in the discussion (highlighted in red) regarding radiomics and added the suggested very interesting articles in the reference list.

Reviewer 3 Report

Authors should be congratulated for their work. Paper is well written, clear, and concise. Methodology is correct. Further validation is needed in external series as reported in the conclusion section. Besides, there is only a minor concern regarding the rate of false positive cases regarding AI vs Reading respect to Reading vs Reading. It seems AI was associated with a significantly higher FP rate. This finding merits further discussion because it may impact prognosis and deviate from the most appropriate clinical path

Author Response

We would like to thank the reviewer for the thorough reading of this manuscript and for the constructive suggestions. We have added a paragraph in the discussion (highlighted in red in the revision) regarding the number of false positives and false negatives.

Round 2

Reviewer 2 Report

The authors answered all comments and suggestions.